# Advances in analytical approaches for background parenchymal enhancement in predicting breast tumor response to neoadjuvant chemotherapy: A systematic review

**Julius Thomas**[1*¤a], **Lucas Malla**[2¤b], **Benard Shibwabo**[1¤a]

**1** School of Computing and Engineering Sciences, Strathmore University, Nairobi, Kenya, **2** London School of Hygiene & Tropical Medicine, London, United Kingdom

¤a Current Address: Karen Ole Sangale Rd, off Langata Road, in Madaraka Estate, Nairobi, Kenya.
¤b Current Address: Keppel St, London WC1E 7HT, United Kingdom.
* thomas.oduor815@strathmore.edu

## Abstract

### Background

Breast cancer (BC) continues to pose a substantial global health concern, necessitating continuous advancements in therapeutic approaches. Neoadjuvant chemotherapy (NAC) has gained prominence as a key therapeutic strategy, and there is growing interest in the predictive utility of Background Parenchymal Enhancement (BPE) in evaluating the response of breast tumors to NAC. However, the analysis of BPE as a predictive bio-marker, along with the techniques used to model BPE changes for accurate and timely predictions of treatment response presents several obstacles. This systematic review aims to thoroughly investigate recent advancements in the analytical methodologies for BPE analysis, and to evaluate their reliability and effectiveness in predicting breast tumor response to NAC, ultimately contributing to the development of personalized and effective therapeutic strategies.

### Methods

A comprehensive and structured literature search was conducted across key electronic databases, including Cochrane Database of Systematic Reviews, Google Scholar, PubMed, and IEEE Xplore covering articles published up to May 10, 2024. The inclusion criteria targeted studies focusing on breast cancer cohorts treated with NAC, involving both pre-treatment and at least one post-treatment breast dynamic contrast-enhanced Magnetic Resonance Imaging (DCE-MRI) scan, and analyzing BPE utility in predicting breast tumor response to NAC. Methodological quality assessment and data extraction were performed to synthesize findings and identify commonalities and differences among various BPE analytical approaches.

**Data availability statement:** All relevant data are contained within the paper and its Supporting Information files.

**Funding:** This research was funded by the Deutscher Akademischer Austauschdienst (DAAD) scholarship under the 2020/2023 In-Country/In-Region Scholarship Programme in Eastern Africa - Kenya (grant 91601895). However, the funders had no involvement in the study design, data collection and analysis, decision to publish, or manuscript preparation.

**Competing interests:** The authors have declared that no competing interests exist.

## Results

The search yielded a total of 882 records. After meticulous screening, 78 eligible records were identified, with 13 studies ultimately meeting the inclusion criteria for the systematic review. Analysis of the literature revealed a significant evolution in BPE analysis, from early studies focusing on single time-point BPE analysis to more recent studies adopting longitudinal BPE analysis. The review uncovered several gaps that compromise the accuracy and timeliness of existing longitudinal BPE analysis methods, such as missing data across multiple imaging time points, manual segmentation of the whole-breast region of interest, and over reliance on traditional statistical methods like logistic regression for modeling BPE and pathological complete response (pCR).

## Conclusion

This review provides a thorough examination of current advancements in analytical approaches for BPE analysis in predicting breast tumor response to NAC. The shift towards longitudinal BPE analysis has highlighted significant gaps, suggesting the need for alternative analytical techniques, particularly in the realm of artificial intelligence (AI). Future longitudinal BPE research work should focus on standardization in longitudinal BPE measurement and analysis, through integration of deep learning-based approaches for automated tumor segmentation, and implementation of advanced AI technique that can better accommodate varied breast tumor responses, non-linear relationships and complex temporal dynamics in BPE datasets, while also handling missing data more effectively. Such integration could lead to more precise and timely predictions of breast tumor responses to NAC, thereby enhancing personalized and effective breast cancer treatment strategies.

## Introduction

Breast cancer (BC) is one of the biggest global health burdens, affecting numerous women, with the incidence and mortality rates rising each year [1–3]. The International Agency for Research on Cancer (IARC) attributes late and sometimes inaccurate diagnosis and ineffective treatment to higher cancer mortality rates [4]. While significant advances in diagnostic techniques have improved outcomes, breast cancer treatment continues to present significant clinical challenges.

Neoadjuvant chemotherapy (NAC) has gained prominence as the primary therapeutic approach due to its potential to downstage tumors and improve treatment outcomes [5]. The preoperative administration of NAC to locally advanced or early-stage breast tumors has been shown to have beneficial effects [6,7]. NAC not only renders previously inoperable tumors operable among breast cancer patients but also increases the likelihood of achieving a pathological complete response (pCR) [6,8,9].

Also, the administration of NAC can facilitate the prediction of tumor response, enabling the tracking and prediction of patients' responses to treatments [10–12]. Consequently, non-responders can discontinue ineffective and toxic treatments, reducing unnecessary expenses and switching to an effective regimen, thereby avoiding delayed surgery or disease progression [13]. Ultimately, by predicting a patient's response to NAC treatment, oncologists can make well-informed decisions regarding the most appropriate treatment plan for each individual patient.

Due to the intratumor heterogeneity of BC and its microenvironments [14], patients' responses to NAC vary significantly during treatment. While some experience considerable tumor shrinkage at the initial stages of treatment, others may show limited response or even complete resistance, and vice vasa. Over time, a small to a significant proportion of patients (6%–45%) achieved pCR [12,15–17], while approximately 5% of patients showed disease progression [18]. Precise and timely prediction of individuals who are unlikely to respond to this treatment strategy is essential for personalized effective therapeutic approach, but still remains a clinical challenge. Therefore, it is difficult to timely tailor the correct NAC treatment regimens to individual patients' who need them the most for optimal outcomes.

In contemporary clinical oncology, analyzing and predicting how a patient will respond to NAC presents a notable challenge. The current approach primarily depends on the Response Evaluation Criteria in Solid Tumors (RECIST) criteria [19], which focuses on measuring how the size of tumor progressively changes with treatment, an evaluation typically conducted after the patient has completed or undergone several cycles of NAC [20].

Given that NAC is often employed to shrink tumors, tumors may not immediately respond to treatment by simply shrinking in size; their functional characteristics and cellular activity can change significantly before any noticeable reduction in size. These changes may constitute decreased metabolic activity, changes in blood flow and composition of cells, and modifications in cellular proliferation rates. Such functional changes, which are essential aspects of the tumor's response to treatment, tend to occur earlier than changes in size [20].

Unfortunately, the current reliance on RECIST, which primarily considers size changes, often overlooks these early functional responses. Waiting until changes in tumor size become noticeable before evaluating the patient's response to NAC to switch to a more effective treatment regimen if the current one is proving ineffective, could potentially compromise the patient's outcome [21]. Consequently, non-responders may continue to endure costly, toxic treatments and experience delays in surgery and disease progression [22,23]

The integration of functional biomarkers in the assessment and prediction of tumor response to NAC could significantly improve the outcome of patients undergoing NAC [12,20,24]. These biomarkers, which include genetic, molecular, metabolic, or imaging-based parameters, can reflect the behavior of tumor and its microenvironment in response to treatment [25,26].

In recent years, Breast Magnetic Resonance Imaging (MRI) has emerged as an essential tool in the management of treatment of breast cancer. The MRI provides high-resolution images that offer detailed anatomical and functional information about breast tissues, aiding in early cancer detection and treatment approaches planning. Within the realm of breast cancer and its associated MRI biomarkers, Background Parenchymal Enhancement (BPE) has garnered growing interest as a more precise imaging characteristic capable of enhancing the prediction of how breast cancer responds to NAC [27–30]. BPE, a dynamic and heterogeneous enhancement pattern observed in the surrounding fibroglandular tissue on contrast-enhanced MRI of the breast [31], has been suggested to hold valuable information regarding tumor microenvironment, angiogenesis and tumor response to therapy.

Analyzing changes in BPE to predict how breast tumor responds to NAC have demonstrated encouraging outcomes [32–38]. However, the significant variability in NAC response stemming from diverse tumor characteristics (such as size, stage, progesterone and estrogen receptors, HER2 status) [39], to NAC protocols (such as regimen type, treatment cycles) [40,41], as well as molecular subtype variations among patients, leads to dynamic changes in BPE during NAC.

The approaches and techniques utilized in analyzing changes in BPE during NAC administration play a crucial role in the accuracy and timeliness of subsequent BPE models used to

predict breast tumor responses to NAC. In contemporary clinical practice, while segmentation methods for fibroglandular tissue (FGT) in BPE quantification have improved, the commonly used semi-automated or automated proprietary techniques for region of interest (ROI) delineation often involve preliminary manual steps. These can lead to inaccuracies in defining the ROI, potentially underestimating or overestimating BPE, which affects both accuracy and reproducibility. Additionally, longitudinal monitoring of BPE changes at various neoadjuvant chemotherapy (NAC) stages faces significant challenges. Inconsistent follow-up schedules or patient non-compliance frequently result in missing MRI measurements after each NAC cycle, which, in turn, excludes these patients from BPE change analysis, limiting sample size and reducing statistical power and generalizability [42, 43].

Moreover, BPE responses to NAC vary widely across cancer subtypes, with some patients responding early, intermediate, others later, and some showing no response. Despite this variability, many studies continue to rely on traditional statistical models, such as logistic regression, to analyze BPE and predict pathological complete response. These models typically assume independence among observations [44], which may overlook the intricate temporal dependencies in longitudinal BPE data. Traditional methods also often assume linear relationships that may fail to capture the complex, nonlinear dynamics inherent in longitudinal BPE, ultimately affecting both the robustness, timeliness and accuracy of models used to predict tumor response to NAC.

Consequently, this study provides a critical review of analytical methodologies used in assessing BPE changes for predicting breast tumor response to NAC, with the primary objectives being: 1) to identify prevailing analytical approaches and methodologies utilized in BPE change analysis; 2) to discuss the advancement in the techniques, and the limitations associated with the identified approaches and methodologies; and 3) to propose recommendations for enhancing the shortcomings in existing analytical approaches and methodologies. These improvements aim to facilitate the development of more robust personalized treatment strategies, enabling the precise and timely identification of non-responders and the optimization of therapeutic approaches for better patient outcomes.

## Materials and methods

The guidelines employed for study searches, selection, and subsequent review procedures adhered to established protocols commonly employed in conducting systematic review studies [45]. Furthermore, we integrated pre-existing PRISMA (Preferred Reporting Items for Systematic Reviews and Meta-Analyses) checklist (S1 File: PRISMA checklist), for reporting reviews outcomes to systematically document the review processes [46]. Our study protocol was registered with the International Prospective Register of Systematic Reviews (PROSPERO) on September 8, 2024, under protocol number CRD42015026904 (accessible at http://www.crd.york.ac.uk/PROSPERO/).

### Eligibility criteria

Inclusion criteria encompassed studies meeting the subsequent conditions: 1) Investigation involving breast cancer cohorts who underwent both pre-treatment and at least one post-treatment breast dynamic contrast-enhanced Magnetic Resonance Imaging (DCE-MRI); 2) Involving patients treated with Neoadjuvant Chemotherapy (NAC); 3) Focused on analyzing changes in background parenchymal enhancement (BPE) across various treatment time-points; 4) Incorporating BPE assessment, either qualitative or quantitative, both before and after treatment initiation; 5) Utilization of breast DCE-MRI scans for BPE measurement to predict tumor response to NAC; 6) Clearly defined patient responses to NAC categorized as

responders or non-responders; 8) Peer-reviewed research articles, review papers, and systematic reviews published in scholarly journals; 9) Publication in any language, as long as it meets the aforementioned criteria.

Conversely, exclusion criteria comprised: 1) Studies not related to the analysis of BPE changes or breast tumor response to NAC; 2) Duplicate or overlapping datasets; 3) Sources lacking peer-review scrutiny, including meta-analyses, conference abstracts, editorials, letters, technical reports, and opinion pieces.

## Study search

In order to identify eligible studies for inclusion, we conducted a comprehensive and structured literature search across key databases such as Cochrane Database of Systematic Reviews, Google Scholar, PubMed, and IEEE Xplore covering articles published up to May 10, 2024. Additionally, we used EndNoteX7 bibliography tool to manually retrieve additional eligible articles and collate the final search results from searched reference lists.

Based on PICO (Problem, Intervention, Comparison, and Outcome) framework [47] as a guideline, we devised a set of search terms, encompassing "background parenchymal enhancement," "breast tumor response," "neoadjuvant chemotherapy," and synonyms relating to "breast," (S2 File: Search Strategy), to systematically sift through databases and pinpoint articles potentially pertinent to our review. We employed Boolean operators (AND, OR) to effectively combine these keywords, thereby formulating a comprehensive search strategy outlined below:

('Breast tumor' OR 'breast cancer') AND ('background parenchymal enhancement' OR 'parenchymal enhancement' OR 'BPE') AND ('neoadjuvant chemotherapy' OR 'NAC') OR ('BPE analysis' OR 'background parenchymal enhancement changes' OR 'BPE changes' OR 'BPE assess') OR ('Dynamic contrast-enhanced MRI" OR 'DCE-MRI') AND ('predict*') AND ('response')

The process entailed structuring of the study objectives and search terms around four key components of the PICO framework, as outlined below:

1. Problem/Population (P)▪ Population: Studies related to patients with breast cancer whose background parenchymal enhancement was measured at different time points during neoadjuvant chemotherapy ('Breast tumor' OR 'breast cancer') AND ('background parenchymal enhancement' OR 'parenchymal enhancement' OR 'BPE') AND ('neoadjuvant chemotherapy' OR 'NAC').

   ▪ Problem: Studies related to inaccurate or untimely prediction of breast tumor response to NAC ('predict*').

2. Intervention (I): Studies related to analytical approaches or methods for assessing BPE changes ('BPE analysis' OR 'background parenchymal enhancement changes OR 'BPE changes' OR 'BPE assess') OR ('Dynamic contrast-enhanced MRI' OR 'DCE-MRI').

3. Comparison (C): Given the study objectives, no analogous investigations were identified to serve as a benchmark.

4. Outcome (O): Studies related to BPE analytical approaches, advancement in BPE analysis, limitations in the current analytical methodologies, and strategies to address identified limitations.

In light of the structural disparities among literature databases, we have meticulously documented tailored search terms for each database utilized. These customized search phrases can be found in S3 File: Database search terms.

## Study selection

Preceding the screening of articles and abstracts, a standardized screening protocol was developed by two reviewers (JT and LM). The study selection process commenced with independent evaluations of the titles and abstracts by both reviewers, guided by predefined inclusion and exclusion criteria to ensure the relevance of the selected articles. Following this initial phase, the two reviewers collaboratively scrutinized the full texts of the chosen articles to assess their eligibility. Any discrepancies were amicably addressed through deliberation, culminating in a mutually agreed consensus between both reviewers.

## Data extraction

In line with our research objectives, we devised a structured data extraction form (S4 File: Data extraction form), which two reviewers (JT and LM) used to independently extract data (if available) from the full-texts of eligible studies. This process aimed to transform the extracted information into a structured format.

The following data variables guided the type of data extracted from the eligible studies: study attributes (such as first author, publication year, and study design), characteristics of the study population (including sample size, and cancer subtype), techniques employed for BPE analysis (covering BPE analysis approach, methods for BPE change analysis, DCE-MRI examination time points, methods for region of interest (ROI) segmentation, BPE assessment techniques, and BPE quantification methods), tumor response phase, nature of data components (whether BPE only, or additional features as well), and technique for model development.

Following data extraction, to ensure accuracy and consistency, extracted data underwent comparative analysis between the two reviewers. Any disparities were addressed through constructive dialogue. In instances where consensus could not be reached, a seasoned clinical expert (possessing over 20 years of expertise in breast imaging) was consulted to arbitrate and resolve any discrepancies.

## Data analysis

The extracted data underwent meticulous analysis to address the objectives of our systematic literature review. Leveraging on the extracted data, we conducted comprehensive analysis and narrow findings to the selection process of studies, as well as the characteristics therein encompassing participants, methodologies, interventions, and outcomes. These insights were presented through descriptive statistical measures including frequencies, proportions, and visual aids such as tables, graphs, and charts, using R statistical programming language version 4.4.2.

## Quality assessment

To accurately interpret the evidence gathered through the selected studies, we rigorously evaluated the risk of bias (ROB) inherent in each study. This evaluation was conducted independently by two reviewers (JT, LM) using a modified Quality Assessment of Studies of Diagnostic Accuracy Studies (QUADAS-2) tool. The assessment process involved grading each study across four key domains, including patient selection, index tests, reference standard and flow and timing [48,49].

Each reviewer separately assessed the quality of the selected studies using the QUADAS-2 tool, meticulously documenting supporting details and rationale for their assessments of ROB across the various domains (categorized as low, high, or unclear). Any discrepancies between reviewers were resolved through discussion until a consensus was reached. In cases where consensus proved elusive, a third reviewer (BS) acted as an impartial adjudicator to facilitate

resolution. The quality assessment is crucial, as it enables identification of any study findings deviations from the truth as a result of flaws in methods used, limitations in the design, or the analysis approach employed.

## Results

### Study selection

A comprehensive search across PubMed (113 records), Google Scholar (110 records), IEEE Xplore (11 records), and the Cochrane Database of Systematic Reviews (1) yielded a total of 882 records. Following meticulous screening to exclude studies unrelated to breast cancer, those focusing on different cancer types, and duplicates, the titles and abstracts of 271 records underwent scrutiny, resulting in the identification of 78 eligible records for inclusion. Subsequent thorough examination of the full texts against predefined selection criteria led to the exclusion of 35 studies due to various reasons: absence of BPE analysis (n = 35), irrelevance to response prediction (n = 15), review or commentary articles (n = 5), conference papers (n = 4), redundancy (n = 2), utilization of radiomic analysis (n = 2), non-DCE-MRI (n = 1), and not pertaining (NAC) (n = 1). Ultimately, thirteen studies met the inclusion criteria and were incorporated into the systematic review. The flow of study selection is visually depicted in S1 Fig, adhering to the Preferred Reporting Items for Systematic Reviews and Meta-Analyses (PRISMA) guidelines [46], illustrating the distinct screening phases.

### Risk of bias

In evaluating the risk of bias in studies analyzing change in BPE to predict tumor response to NAC, we found that the patient selection domain consistently showed a low risk of bias. This was attributed to the detailed descriptions of patient enrollment processes and the appropriate articulation of inclusion and exclusion criteria across all studies. However, the domain evaluating the index test presented an unclear risk of bias in more than half of the studies reviewed [28,36,38,50–53]. This uncertainty arose because some studies did not specify whether the evaluators were blinded to additional information during the qualitative assessment of BPE or whether BPE assessments were conducted independently of knowledge of pCR outcomes (Table 1).

The reference standard domain exhibited a low risk of bias across all studies, as they consistently defined pCR as the reference standard for determining tumor response to chemotherapy. Conversely, the flow and timing domain showed an unclear risk of bias in seven studies [32,51–55,36], which did not provide precise timing details for the breast MRI scans used to subsequently measure BPE in relation to NAC cycles, often resorting to general terms such as "before NAC" or "after NAC". Notably, the study by Chen et al. [36] displayed a high risk of bias in this domain as it showcased inconsistent timing of MRI follow-up examinations for BPE measurements, which could impact on the accuracy of BPE assessment. When examining the applicability concerns of the selected studies, all studies demonstrated a low risk of bias across all domains, as summarized in Table 1 and illustrated in S2 Fig.

### General characteristics of selected studies

Nearly all the studies included in this review were retrospective, featuring both single-center and multicenter designs. Notably, Arasu et al. [51], conducted the only prospective, multicenter study. Sample sizes varied significantly, from 46 to 990 breast cancer patients, showcasing the diverse scope of research in this field.

The focus on BPE alone as a functional biomarker to predict tumor response to NAC was predominant across most of the studies reviewed [32,36,38,53–55,56,57], with only one-third [28,50,51,58] also featuring additional imaging biomarkers, such as functional tumor volume, breast tumor size, longest diameter, sphericity and fibroglandular tissue measurements, aiming to enhance the predictive accuracy for NAC response.

Remarkably, the studies from the review applied the analysis of BPE as a predictor of breast tumor response across various breast cancer subtypes and stages, with stage II and II dominating the cancer stages [28,51,57,58], and only a study by La Forgia et al. [28] reporting on stage I and II. Regarding the subtypes, human epidermal growth factor receptor 2 (HER2) was the most predominant [28,32,51,55,57,58], followed by invasive ductal and lobular carcinomas [36,50,54]. Other subtypes included hormone receptor (HR) positive and negative [51,53,57,58], and luminal A, luminal B, triple negative, and triple positive as noted in Forgia et al. [28].

Additionally, patient exclusions due to missing MRI timepoints, as captured in Table 2, caused by irregular follow-up schedules, non-compliance, or unknown reasons, ranged from 1 to 255 patients, underscoring the challenges associated with MRI data collection.

## BPE analysis techniques

**BPE analysis approach.**  To analyze BPE as a predictive biomarker for breast tumor response to NAC, a review of the literature revealed that about one-third of the studies [32,54–55,38] adopted a single time-point BPE analysis approach, while approximately two-thirds [36,50–53,58,57] employed longitudinal BPE analysis approach. Studies employing single time-point BPE analysis typically assess BPE at baseline before NAC initiation and again after completing NAC or following a specified number of NAC cycles. Conversely, longitudinal BPE analysis studies analyze BPE changes from baseline through multiple time points across different phases of NAC treatment, including early, mid-treatment, and post-treatment stages as shown in Table 3.

**BPE assessment and quantification.**  In earlier studies, qualitative assessments were predominantly used to measure BPE, involving visual comparison of BPE changes based on Breast Imaging-Reporting and Data System (BI-RADS) criteria [31] between two time points, particularly in single time-point BPE analysis studies [32,38,54–55]. In contrast, majority of the recent longitudinal BPE analysis studies have largely incorporated quantitative approaches [51–53,57,58] using advanced segmentation techniques, such as deep learning-based segmentation (nnU-Net) in Huang et al. [57], automated breast segmentation [53,57,58] and semi-automated segmentation in Rella et al. [52]. An exception to this trend is the study by Arasu et al. [51], which utilized a whole breast manual segmentation method (refer to Table 3).

**Tumor to NAC response phase.**  The timing at which a significant association between changes in BPE and pCR is first detected during NAC varied significantly across reviewed studies. Most studies utilizing single time-point BPE analysis reported detecting significant changes in BPE and its association with pCR during the late phase of NAC [32,38,54] or at an intermediate stage of treatment [55]. However, Forgia et al [28] uniquely reported a significant association during the early phase of NAC. Among the longitudinal BPE analysis studies, while several studies [28,36,50–52,57] first detected a significant association between BPE reduction and pCR in early phase of NAC treatment, a study by Onishi et al. [53] observed this at varied treatment phases, depending on breast cancer subtype, as showcased in Table 3.

**Model development techniques.**  Various statistical approaches have been utilized across different studies to determine whether changes in BPE can serve as predictors for tumor response to NAC. While statistical methods such as logistic regression was applied by nearly half of the reviewed studies [50,51,57,58] to predict tumor response to NAC, several other

studies [28,32,36,38,52–55] opted for more straightforward statistical tests to achieve similar predictive aims (refer to Table 3).

## Discussion

BPE represents a promising functional biomarker with the potential to transform the prediction of breast tumor response to NAC, enabling precise and timely personalized treatments. However, the process of coming up with BPE to serve as a predictive biomarker, along with the methods employed to model BPE for the accurate and prompt identification of non-responders, responders, and disease progressors to NAC, is fraught with several challenges. We undertook a comprehensive review of the current literature to examine advancements in analytical techniques used to assess BPE changes and their effectiveness in predicting breast tumor response to NAC.

Overall, approximately two-thirds of the reviewed literature sources [32,36,38,52–55,56,57] focused on BPE analysis as a functional biomarker for predicting tumor response to NAC. This underscores BPE's potential as a standalone biomarker providing critical insights for making informed decisions regarding breast tumor response to NAC. Moreover, the application of BPE analysis across various breast cancer types and stages highlights its utility in serving a broad spectrum of the breast cancer population. This capability enhances the precision of predicting treatment responses, thereby supporting the development of more individualized treatment plans.

Based on the analysis of published literature, it is evident that the approach to BPE analysis has evolved significantly. Initially, BPE studies predominantly focused on single time-point BPE analysis, as highlighted by earlier research [32,38,54–55]. In contrast, more recent studies have shifted towards longitudinal BPE analysis [36,50–53,57,58]. While a single pre-and-post BPE analysis has demonstrated positive outcomes in using BPE as a functional biomarker for predicting breast tumor response to NAC [32,38,54–55], the analysis process comes with certain inherent limitations that may hinder timely, precise, and personalized prediction of tumor response to NAC. One major constraint is its static nature, capturing change in BPE data at single time-point. Given the heterogeneous nature of breast tumor responses to NAC, a single time-point analysis may not adequately reflect the dynamic changes in BPE that occur during various treatment phases. Tumors and their microenvironments are inherently dynamic, exhibiting varied responses at different treatment stages, and a single evaluation might overlook crucial temporal changes [53,59,60].

Additionally, relying on a single post-treatment BPE analysis delays the detection of early indicators of tumor response, forcing clinicians to wait until after treatment completion to assess a one-time change. This delay can prevent timely adjustments to effective treatment plans, potentially compromising patient outcomes. To overcome the inherent limitations in single time-point BPE analysis and enhance breast cancer treatment strategies, a significant number of recent studies have adopted a longitudinal approach [36,50–53,57,58] with promising outcomes. By tracking BPE changes at multiple NAC treatment intervals, researchers have gained critical insights into the dynamic nature of breast tumors and their surrounding microenvironment throughout the treatment process [60]. This longitudinal monitoring captures subtle variations over different time points, facilitating more accurate and timely predictions of treatment response, thereby enabling more personalized therapeutic strategies.

While there has been a recent shift in research towards longitudinal approaches for analyzing BPE, it is noteworthy that some studies, such as La Forgia et al. [28], continue to utilize qualitative methods for BPE assessment. This qualitative approach involves experienced radiologists visually evaluating BPE using the BI-RADS lexicon, categorizing it as marked,

moderate, mild, or minimal [31]. However, such methods not only reduce the accuracy of BPE measurements due to the introduction of subjective bias but also exacerbate issues of inter- and intra-observer variability, thus posing significant challenges in terms of reliability and reproducibility [28,61–63]. These inherent shortcomings compromise both the resultant BPE data, and any predictive model developed thereof.

In response to these identified limitations, a growing body of recent longitudinal studies [36,50–53,57,58] has increasingly embraced diverse methodologies aimed at robustly quantifying BPE. This strategic shift offers the advantage of minimizing variability inherent in human interpretation by radiologists, while also enabling quantification of subtle functional changes within breast tissues not easily detected through conventional visual assessments [64]. The quantitative evaluation of BPE typically encompasses three pivotal steps: the initial segmentation of the entire breast volume from a pre-contrast image, followed by the segmentation of fibroglandular tissue (FGT) within the delineated breast volume, culminating in the precise quantification of BPE [65,66]. The process of whole-breast segmentation involves the manual delineation of the fibroglandular tissue boundary [20] or alternatively, the utilization of automated [67] or semi-automated techniques for tumor segmentation [52,68,69].

Although the recommendation for embracing fully automated methodologies for quantifying BPE across various time point is clear, a few investigations opted for either manual whole breast segmentation [51] or semi-automated breast segmentation [52]. While these methods yielded more reliable outcomes compared to qualitative BPE assessments, as demonstrated in the study by Rella et al. [52], they were marred by some level of manual preliminary image segmentation of the region of interest (ROI), compromising the accuracy and reproducibility of BPE measurement. Consequently, it was advised that fully automated segmentation techniques, particularly those leveraging artificial intelligence, be prioritized instead, to ensure enhanced accuracy and reproducibility in BPE measurement.

Given the critical importance of accurate segmentation of FGT for precise quantification of BPE, numerous longitudinal studies in the literature have made strides in advancing methods for segmenting the ROI, and have employed various automated techniques to enhance accuracy and reproducibility. For instance, Huang et al. [57] utilized a deep learning-based segmentation approach using nnU-Net, while other researchers [53,57,58] employed diverse automated methods. However, a study by Onishi et al. [53] highlighted inaccuracies in BPE quantification due to the utilization of proprietary tools for ROI delineation, which overestimated breast tumor regions, incorporating benign tumors. To address these challenges, there is a compelling need to transition towards alternative deep learning-based approaches. Embracing such methodologies holds the promise of a reproducible, more precise, and automated tumor segmentation process [67,70–72]. The shift could significantly enhance the accuracy of BPE quantification, thereby mitigating the drawbacks associated with current quantification methodologies. Consistent with this notion, studies comparing manual and existing algorithms for whole breast and FGT segmentation have demonstrated that deep learning-based approaches such as U-net methods outperform existing techniques [65]. Furthermore, these methods not only offer reproducibility but also enable more accurate BPE quantification and facilitate the development of improved predictive frameworks [73].

Despite the advancement in refining techniques to bolster the precision and promptness of longitudinal BPE analysis studies in predicting how breast tumor responds to NAC, various constraints have surfaced. Primarily, while all of these studies have employed conventional statistical methodologies, such as logistic regression, to model changes in BPE over multiple time points in predicting pathologic complete response, inherent limitations persist. Logistic regression hinges on the assumption of independence among observations [43,44,74], potentially overlooking the intricate temporal dependencies inherent in longitudinal BPE data.

Moreover, these traditional statistical approaches assume linear relationships, which may not fully capture diverse, intricate, complex, and non-linear dynamics present in longitudinal BPE datasets.

Longitudinal BP monitoring across multiple time-points during NAC treatment has facilitated the timely identification of individualized responses during different phases of treatment. This is imperative not only for the early identification of non-responders and responders, but also for prompt decisions regarding treatment adjustments [51,52,57,58]. Various patterns of tumor response phases have been observed across the longitudinal studies scrutinized. While the majority of these studies first detected a significant association between BPE reduction and pCR in early phase of NAC treatment, a study by Onishi et al. [53] documented this phenomenon at varied phases of treatment, depending on breast cancer subtype. Breast tumors exhibit heterogeneity [53,59,60], and thus will respond variedly to NAC. The varied nature of tumor responses at different times, with some responding swiftly, others exhibiting delayed responses, or even showing resistance, poses a complexity that traditional statistical approaches may struggle to grasp effectively. Adopting these conventional techniques, unfortunately, may fail to capture these early response indicators, thereby delaying potential treatment adjustments or resulting in inaccurate predictions of treatment outcomes.

Several studies reported missing BPE data points at one or more MRI time points during the course of NAC treatment, and that any patient who had one or more MRI data points missing at any time interval was excluded from BPE analysis. This is because traditional statistical methods may struggle with the missing data points [42] as a result of irregular follow-up schedules or patient non-compliance. Imputation methods, when used, might introduce biases, leading to less robust predictions and inaccurate results. A study by You et al. [50] highlighted this limitation, where the unavailability of certain observations due to irregular follow-up during the 8th NAC cycle led to the exclusion of the entire 8th follow-up MRI time point data from the logistic model.

Excluding patients with missing data may pose challenges to the integrity of study outcomes, primarily by reducing the sample size, and subsequently diminishing statistical power of the study and limiting the generalizability of its findings [75–77]. Moreover, longitudinal data inherently contains complex temporal dependencies, meaning that observations at different time points are related. Traditional methods like logistic regression assume independence among observations and may overlook these dependencies. Excluding data points disrupts the continuity and may ignore meaningful temporal patterns in BPE changes, resulting in less robust and inaccurate models for forecasting breast tumor response to NAC. Advanced statistical techniques, such as mixed-effects models or machine learning techniques, can handle missing data more effectively without excluding patients [76,78–81]. In a similar study by Jerez et al. [82] to use all the available data and not discard records with missing values, it was reported that imputation methods based on machine learning algorithms outperformed imputation statistical methods in the prediction of patient outcome, and led to a significant enhancement of prognosis accuracy. Overall, these methods can model the temporal correlation and handle non-linear relationships better, providing more accurate and robust predictions.

While this review highlights significant advancements in BPE analytical approaches for predicting breast tumor response to NAC, it is essential to acknowledge the limitations of the included evidence. Firstly, there is no standardized method for measuring and quantifying BPE, leading to variability in assessment across studies. Differences in MRI protocols, BPE segmentation methods, and analysis approaches can affect the accuracy and reproducibility of BPE measurements and complicate the analysis of the relationship between BPE and tumor response. The adoption of advanced analytical techniques, such as deep learning and other

AI-based methods, holds promise for standardizing protocols for longitudinal BPE quantification and analysis. This could enhance the reproducibility and accuracy of BPE assessments, facilitating standardized comparisons across different studies.

Additionally, BPE is influenced by several factors, including hormonal status, age, and genetic predispositions. The reviewed studies may not have fully accounted for these confounding variables, potentially leading to inaccuracies when comparing outcomes. For personalized care and treatment strategies, future research should advocate for patient-centered approaches that consider individual patient factors when analyzing BPE variability and NAC response.

Long-term follow-up data are crucial for understanding the predictive value of BPE. However, many studies lack sufficient longitudinal data, making it difficult to assess the long-term outcomes of NAC and the role of BPE in predicting these outcomes. While BPE is an important imaging biomarker, the prediction of NAC response is multifactorial and may benefit from integrating additional data, such as genetic, histopathological, and other imaging biomarkers. Studies focusing solely on BPE may overlook the potential benefits of a more comprehensive, multimodal approach.

The review process itself also has limitations. The included studies vary widely in terms of design, patient populations, imaging techniques, and analytical methods. This heterogeneity can make it challenging to compare results directly and draw definitive conclusions.

## Conclusions

The advancements in analytical approaches for assessing BPE underscore its potential as a robust functional biomarker for predicting breast tumor response to NAC. The shift towards longitudinal BPE analysis approach has exposed significant gaps, particularly in varied nature of tumor responses at different treatment times and the challenges in BPE quantification, especially during whole-breast region of interest segmentation. Traditional statistical methods, such as logistic regression, are often inadequate for modeling longitudinal changes in BPE due to their inability to handle complex temporal dependencies within BPE and effectively manage missing BPE data in one or more time-points without excluding patients with incomplete datasets.

In light of these gaps, there is a clear call in the literature to consider adopting alternative analytical techniques, particularly in the realm of artificial intelligence (AI). Future longitudinal BPE research work should focus on standardization in longitudinal BPE measurement and analysis, through integration of deep learning-based approaches for automated tumor segmentation, and implementation of advanced AI technique that can better accommodate varied breast tumor responses, non-linear relationships and complex temporal dynamics in BPE data, while also managing missing data more effectively. Additionally, consideration of patient-specific factors in longitudinal BPE analysis, and further integration of multimodal data can enhance the robustness of evidence and improve our understanding of BPE's role in predicting NAC response.

By embracing these innovative approaches, researchers can achieve more precise analyses and make more robust and timely predictions in longitudinal BPE studies, ultimately enhancing personalized treatment strategies for breast cancer patients.

## Supporting information

**S1 File. PRISMA checklist.** PRISMA checklist.
(DOCX)

**S2 File. Search strategy.** Search strategy for the systematic review of advances in analytical approaches for background parenchymal enhancement in predicting breast tumor response to neoadjuvant chemotherapy.
(DOCX)

**S3 File. Database search terms.** Search Terms as per Databases during Study Search.
(DOCX)

**S4 File. Data extraction form.** Data extraction form for the systematic review of advances in analytical approaches for background parenchymal enhancement in predicting breast tumor response to neoadjuvant chemotherapy.
(DOCX)

**S5 File. Table of all studies.** List of all studies reviewed.
(XLSX)

**S6 File. Table of extracted data.** Extracted data.
(XLSX)

**S7 File. Table of grouped bar graph of risk of bias and applicability concerns.** Risk of Bias and Applicability Concerns Data.
(XLSX)

**S1 Table. Risk of bias assessment and applicability concerns.** The overall risk of bias assessment and applicability concerns for the selected studies derived using the revised Quality Assessment of Diagnostic Accuracy Studies (QUADAS-2) tool.
(DOCX)

**S2 Table. Relationship between BPE and NAC Response.** Overview of the studies that evaluated relationship between BPE and NAC response.
(DOCX)

**S3 Table. BPE Analytical Methods.** Overview of the analytical methods utilized for BPE in predicting breast tumor response to neoadjuvant chemotherapy.
(DOCX)

**S1 Fig. PRISMA Flow Diagram.** Flow diagram of preferred reporting items for systematic reviews and meta-analyses (PRISMA).
(TIF)

**S2 Fig. Risk of bias and applicability concerns for the selected studies.** Grouped bar graph of risk of bias and applicability concerns for the selected studies derived using the revised Quality Assessment of Diagnostic Accuracy Studies (QUADAS-2) tool.
(TIF)

## Author contributions

**Conceptualization:** Julius Thomas, Benard Shibwabo.

**Data curation:** Julius Thomas, Lucas Malla.

**Formal analysis:** Julius Thomas, Lucas Malla.

**Funding acquisition:** Julius Thomas, Benard Shibwabo.

**Investigation:** Julius Thomas.

**Methodology:** Julius Thomas.

**Project administration:** Julius Thomas.

**Supervision:** Benard Shibwabo.

**Validation:** Lucas Malla.

**Writing – original draft:** Julius Thomas.

**Writing – review & editing:** Julius Thomas, Benard Shibwabo.

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
