## [Decision Letter · Decision Letter 0]

29 Oct 2024

PONE-D-24-30545Advances in analytical approaches for background parenchymal enhancement in predicting breast tumor response to neoadjuvant chemotherapy: A comprehensive reviewPLOS ONE

Dear Dr. Thomas,

Thank you for submitting your manuscript to PLOS ONE. After careful consideration, we feel that it has merit but does not fully meet PLOS ONE’s publication criteria as it currently stands. Therefore, we invite you to submit a revised version of the manuscript that addresses the points raised during the review process.

We look forward to receiving your revised manuscript.

Kind regards,

Jiang Gui

Academic Editor

PLOS ONE

Journal Requirements:

 Deutscher Akademischer Austauschdienst scholarship (DAAD) under the 2020/2023 In-Country/In-Region Scholarship Programme in Eastern Africa -Kenya (grant 91601895).  

Please respond by return e-mail so that we can amend your financial disclosure and competing interests on your behalf.

Additional Editor Comments:

Dear Authors,

Please the address the minor concerns raised by reviewer 1 regarding on introduction and reference sections. Thanks.

Reviewers' comments:

Reviewer's Responses to Questions

**Comments to the Author**

1. Is the manuscript technically sound, and do the data support the conclusions?

Reviewer #1: Yes

2. Has the statistical analysis been performed appropriately and rigorously? 

Reviewer #1: Yes

3. Have the authors made all data underlying the findings in their manuscript fully available?

Reviewer #1: Yes

4. Is the manuscript presented in an intelligible fashion and written in standard English?

Reviewer #1: Yes

5. Review Comments to the Author

Reviewer #1: This systematic review is important as it synthesizes current knowledge on BPE as a biomarker for predicting breast tumor response to NAC, addressing both advancements and challenges in the field. It contrasts traditional single time-point analyses with longitudinal approaches, emphasizing their advantages in capturing dynamic changes. The review emphasizes the significance of personalized treatment strategies and the integration of various data types, ultimately aiming to enhance patient outcomes through improved predictive models in breast cancer management.

The introduction is well-structured and informative.

The methodology section is robust, comprehensive and well-aligned with PLOS ONE's rigorous standards. Multiple databases (PubMed, Google Scholar, IEEE Xplore, Cochrane) were used and search strategy with Boolean operators according to PICO framework enhances the precision and clarity. The risk of bias was rigorously evaluated by including modified QUADAS-2 tool. The PRISMA checklist, search terms according to PICO framework, and inclusion/exclusion criteria are well-documented, contributing to transparency and reproducibility.

The discussion identifies limitations in current methodologies, including subjective assessments and challenges with missing data. Recommendations for standardizing BPE measurement and incorporating personalized patient factors are made to enhance predictive accuracy. Overall, the review provides valuable insights into the complexities of utilizing BPE in clinical settings, highlighting the need for a multimodal approach in treatment strategies.

Just a few minor suggestions before to proceed publishing:

• While the introduction is logical and cites relevant literature, it would benefit from a more explicit discussion of gaps in the existing research that your review aims to address. (i.e. more on BPE change analysis.) This would strengthen the justification and comprehension of the study.

• Two of the references were not provided as numbers, should be corrected.

6. PLOS authors have the option to publish the peer review history of their article (what does this mean? ). If published, this will include your full peer review and any attached files.

**Do you want your identity to be public for this peer review?** For information about this choice, including consent withdrawal, please see our Privacy Policy .

Reviewer #1: No

---

## [Author Response · Author response to Decision Letter 0]

17 Dec 2024

Response to reviewers

We sincerely thank the editor and reviewers for their valuable time and thoughtful evaluation of our manuscript. To address the feedback provided, we have crafted detailed, line-by-line responses to each comment. For clarity, our responses are highlighted in blue and correspond directly to the comments or questions, which are marked in yellow. Additionally, we have referenced the review decision from the editorial manager and maintained the same format to structure our responses systematically.

INITIAL COMMENTS

PONE-D-24-30545

Advances in analytical approaches for background parenchymal enhancement in predicting breast tumor response to neoadjuvant chemotherapy: A comprehensive review

PLOS ONE

Dear Dr. Thomas,

Thank you for submitting your manuscript to PLOS ONE. After careful consideration, we feel that it has merit but does not fully meet PLOS ONE’s publication criteria as it currently stands. Therefore, we invite you to submit a revised version of the manuscript that addresses the points raised during the review process.

We look forward to receiving your revised manuscript.

Response: Thank you for the constructive feedback and helpful guidance on protocol registration and reproducibility resources. We are pleased to confirm that we have already registered and published our protocol with the International Prospective Register of Systematic Reviews (Protocol Identifier: CRD42024574312) on September 8, 2024 (http://www.crd.york.ac.uk/PROSPERO/).

For your review, we have incorporated this identifier into the manuscript (please refer to Materials and Methods section, page 6).

Kind regards,

Jiang Gui

Academic Editor

PLOS ONE

Journal Requirements:

Response: Thank you for pointing this out. We have carefully reviewed the PLOS ONE style requirements and ensured that our manuscript fully complies, including file naming conventions, bibliography styles, abstract, introduction, and figure formatting. We greatly appreciated the provided style templates, which were invaluable in guiding our revisions.

Deutscher Akademischer Austauschdienst scholarship (DAAD) under the 2020/2023 In-Country/In-Region Scholarship Programme in Eastern Africa -Kenya (grant 91601895).

Please respond by return e-mail so that we can amend your financial disclosure and competing interests on your behalf.

Response: We commend the guidance regarding the financial disclosure! We confirm that the statement is correct as written: "The funders had no role in study design, data collection and analysis, decision to publish, or preparation of the manuscript". Thank you for your assistance in finalizing these details.

Response: Thank you for this important reminder. We have carefully reviewed our reference list to confirm its completeness and accuracy. We verified that none of the cited papers have been retracted; however, we have only updated the two references to Numbered style for uniformity.

Additional Editor Comments:

Dear Authors,

Please the address the minor concerns raised by reviewer 1 regarding on introduction and reference sections. Thanks.

Reviewers' comments:

Reviewer's Responses to Questions

Comments to the Author

1. Is the manuscript technically sound, and do the data support the conclusions?

Reviewer #1: Yes

2. Has the statistical analysis been performed appropriately and rigorously?

Reviewer #1: Yes

3. Have the authors made all data underlying the findings in their manuscript fully available?

The PLOS Data policy requires authors to make all data underlying the findings described in their manuscript fully available without restriction, with rare exception (please refer to the Data Availability Statement in the manuscript PDF file). The data should be provided as part of the manuscript or its supporting information, or deposited to a public repository. For example, in addition to summary statistics, the data points behind means, medians and variance measures should be available. If there are restrictions on publicly sharing data—e.g., participant privacy or use of data from a third party—those must be specified.

Reviewer #1: Yes

4. Is the manuscript presented in an intelligible fashion and written in standard English?

Reviewer #1: Yes

5. Review Comments to the Author

Reviewer #1: This systematic review is important as it synthesizes current knowledge on BPE as a biomarker for predicting breast tumor response to NAC, addressing both advancements and challenges in the field. It contrasts traditional single time-point analyses with longitudinal approaches, emphasizing their advantages in capturing dynamic changes. The review emphasizes the significance of personalized treatment strategies and the integration of various data types, ultimately aiming to enhance patient outcomes through improved predictive models in breast cancer management.

The introduction is well-structured and informative.

The methodology section is robust, comprehensive and well-aligned with PLOS ONE's rigorous standards. Multiple databases (PubMed, Google Scholar, IEEE Xplore, Cochrane) were used and search strategy with Boolean operators according to PICO framework enhances the precision and clarity. The risk of bias was rigorously evaluated by including modified QUADAS-2 tool. The PRISMA checklist, search terms according to PICO framework, and inclusion/exclusion criteria are well-documented, contributing to transparency and reproducibility.

The discussion identifies limitations in current methodologies, including subjective assessments and challenges with missing data. Recommendations for standardizing BPE measurement and incorporating personalized patient factors are made to enhance predictive accuracy. Overall, the review provides valuable insights into the complexities of utilizing BPE in clinical settings, highlighting the need for a multimodal approach in treatment strategies.

Just a few minor suggestions before to proceed publishing:

Comment:

• While the introduction is logical and cites relevant literature, it would benefit from a more explicit discussion of gaps in the existing research that your review aims to address. (i.e. more on BPE change analysis.) This would strengthen the justification and comprehension of the study.

Response: Thank you for your constructive feedback to enhance the clarity and relevance of our work. We appreciate your suggestion and agree that an expanded discussion of the existing gaps in BPE change analysis could better highlight the unique contributions of our review. We have revised the manuscript’s introduction to provide a more detailed analysis of these gaps, specifically focusing on the current limitations in longitudinal BPE change analysis (please refer to Introduction section, page 5 and 6).

Comment:

• Two of the references were not provided as numbers, should be corrected.

Response: Thank you for your careful review. We have now updated the two references to Numbered style and ensured accurate numbering in the revised manuscript version (please see Introduction section, page 4).

6. PLOS authors have the option to publish the peer review history of their article (what does this mean?). If published, this will include your full peer review and any attached files.

Do you want your identity to be public for this peer review? For information about this choice, including consent withdrawal, please see our Privacy Policy.

Reviewer #1: No

Comment:

Response: We have registered with PACE, and all figures are consistent with PLOS guidelines. We downloaded them from PACE and uploaded them as Fig 1 and Fig 2 figures in TIFF format.

ADDITIONAL COMMENTS

PONE-D-24-30545R1

Advances in analytical approaches for background parenchymal enhancement in predicting breast tumor response to neoadjuvant chemotherapy: A comprehensive review

Dear Dr. Thomas,

We've checked your submission and before we can proceed, we need you to address the following issues:

Comment:

1. We note that your Data Availability Statement is currently as follows:

"All relevant data are within the manuscript and its Supporting Information files."

Response: Thank you for bringing this up. We confirm that the minimal dataset required to replicate the results of our study has been uploaded as part of the Supporting Information files accompanying the manuscript. This dataset includes:

• Table of Studies Identified in the Literature Search (refer to ‘Table of all Studies’);

• Table of Extracted Data from the primary research sources for the systematic review (refer to ‘Table of Extracted Data’);

• Table of Risk of Bias and applicability concerns for the selected studies based on revised Quality Assessment of Diagnostic Accuracy Studies (QUADAS-2) tool (refer to ‘Table of Risk of Bias and Quality Assessment’);

• A table of values used to build grouped bar graph of risk of bias and applicability concerns for the selected studies derived using the revised Quality Assessment of Diagnostic Accuracy Studies (QUADAS-2) tool (refer to ‘Table of Grouped Bar Graph of Risk of Bias and Applicability Concerns’);

All relevant data and supporting materials are available within the manuscript and its Supporting Information files, ensuring transparency and compliance with PLOS' data

---

## [Editor Report · Decision Letter 1]

26 Dec 2024

Advances in analytical approaches for background parenchymal enhancement in predicting breast tumor response to neoadjuvant chemotherapy: A systematic review

PONE-D-24-30545R1

Dear Dr. Thomas,

We’re pleased to inform you that your manuscript has been judged scientifically suitable for publication and will be formally accepted for publication once it meets all outstanding technical requirements.

Kind regards,

Jiang Gui

Academic Editor

PLOS ONE
---

## [Editor Report · Acceptance letter]

PONE-D-24-30545R1

PLOS ONE

Dear Dr. Thomas,

I'm pleased to inform you that your manuscript has been deemed suitable for publication in PLOS ONE. Congratulations! Your manuscript is now being handed over to our production team.

Kind regards,

on behalf of

Dr. Jiang Gui

Academic Editor

PLOS ONE